# Antioxidant Response, Phenolic Compounds and Yield of *Solanum tuberosum* Tubers Inoculated with Arbuscular Mycorrhizal Fungi and Growing under Water Stress

**DOI:** 10.3390/plants12244171

**Published:** 2023-12-15

**Authors:** Javiera Nahuelcura, Tiare Ortega, Fabiola Peña, Daniela Berríos, Analía Valdebenito, Boris Contreras, Christian Santander, Pablo Cornejo, Antonieta Ruiz

**Affiliations:** 1Departamento de Ciencias Químicas y Recursos Naturales, Scientific and Technological Bioresource Nucleus BIOREN-UFRO, Universidad de La Frontera, Avda. Francisco Salazar 01145, Temuco 4811230, Chile; jnahuelcurav@gmail.com (J.N.); t.ortega02@ufromail.cl (T.O.); f.pena11@ufromail.cl (F.P.); d.berrios01@ufromail.cl (D.B.); a.valdebenito07@ufromail.cl (A.V.); c.santander01@ufromail.cl (C.S.); 2Programa de Doctorado en Ciencias Agroalimentarias y Medioambiente, Facultad de Ciencias Agropecuarias y Forestales, Universidad de La Frontera, Temuco, Región de la Araucanía, Temuco 4811230, Chile; 3Novaseed Ltda., Loteo Pozo de Ripio s/n, Parque Ivian II, Puerto Varas 5550000, Chile; boriscontreras@novaseed.cl; 4Escuela de Agronomía, Facultad de Ciencias Agronómicas y de los Alimentos, Pontificia Universidad Católica de Valparaíso, Quillota 2260000, Chile; 5Centro Regional de Investigación e Innovación para la Sostenibilidad de la Agricultura y los Territorios Rurales, CERES, La Palma, Quillota 2260000, Chile

**Keywords:** potato, *Claroideoglomus lamellosum*, *Claroideoglomus claroideum*, mycorrhiza, hydroxycinnamic acids

## Abstract

*Solanum tuberosum* (potato) is one of the most common crops worldwide; however, it is sensitive to water stress, which necessitates the identification of alternative tools to improve their production. Here, we evaluated the inoculation of two arbuscular mycorrhizal fungi (AMF) strains, *Claroideoglomus claroideum* (CC), *Claroideoglomus lamellosum* (HMC26), and the MIX (CC + HMC26) in yield and phenolic and antioxidant response using chromatographic and spectroscopic methods in potato crops, at increasing levels of water stress, namely, with 100% (0), 70% (S1), and 40% (S2) soil humidity. Two caffeoylquinic acid isomers were detected and their levels showed a tendency to increase under stress together with the AMF inoculation, reaching up to 19.2 mg kg^−1^ of 5-caffeoylquinic acid and 7.4 mg kg^−1^ of caffeoylquinic acid isomer when CC was inoculated, and potato plants grew at the highest water starvation condition (S2). Regarding antioxidant activities, a differentiated response was detected depending on the AMF strain, highlighting the effect of HMC26 on Trolox equivalent antioxidant capacity (TEAC) method and CC in cupric reducing antioxidant capacity (CUPRAC) method, reaching up to 1.5 μmol g^−1^ of TEAC in plants inoculated with HMC26 and 0.9 μmol g^−1^ of CUPRAC in plants inoculated with CC, both in potato tubers of plants growing under the S2 stress condition. Meanwhile, the use of AMF did not influence the number and biomass of the tubers, but significant changes in the biochemical properties of tubers were observed. The results suggest that specific AMF adaptations to water stress must be considered when inoculation procedures are planned to improve the yield and quality of tubers in potato crops.

## 1. Introduction

Globally, there has been an increase in the prevalence of malnutrition, with approximately 8.4% of the world population having suffered from hunger in 2019, and due to the COVID-19 pandemic, this percentage reached almost 10% in 2020 [1]. There has also been an increase in food insecurity, with 33% of the global population lacking access to sufficient food [1]. Access to nutritious foods, such as cereals, legumes, vegetables, fruits, roots, and tubers, is crucial, with the nutritious potato (*Solanum tuberosum*) being one of the most cultivated and harvested crops in the world after sugar cane, corn, rice, and wheat [1]. Drought generally involves some negative effects on plants, related to the alteration of morphological characteristics especially increasing leaf size and reducing plant and leaf water contents, xylem sap flow, and whole plant hydraulic conductivity [2]. Potato, in addition to having a low fat content, has important nutritional value due to its high content of carbohydrates, high-quality proteins, dietary fiber, minerals, vitamins, and polyphenols [3,4,5].

The optimal temperatures for tuber formation are between 15 and 20 °C, and the tuberization signal can be inhibited at higher temperatures [6]. Under the current scenario of climate change and its effects, potato crops can be severely affected since they are particularly sensitive to increased temperature and water scarcity [7]. Additionally, potato plants have a superficial root system, which makes it difficult for their roots to absorb water in the event of water shortage, causing a decrease in yield of up to 87% in severe cases [8]. It has been reported that drought stress delays the canopy development of plants and, depending on the genotype and the growth stage at which the stress occurs, decreases in early growth and deficient development of the tubers can occur [9,10]. Under stress, reactive oxygen species (ROS) are produced, and in response, there is an increase in the amount of enzymatic and non-enzymatic antioxidants, such as phenolic compounds [11].

It has been reported that under water stress conditions, higher levels of phenolic compounds are synthesized to compensate for the decrease in growth, as has been observed in the roots of *Echinacea purpurea* [12]; moreover, in *Carica papaya*, increases in the levels of phenolic compounds and antioxidant activity have also been observed [13]. This could be of great importance since the presence of phenolic compounds in food has beneficial effects on human health, such as those conferred by the anticancer, analgesic, anti-inflammatory, antimicrobial, and cardioprotective properties of various phenolic compounds [14].

On the other hand, it is well-known that arbuscular mycorrhizal fungi (AMF) form symbiotic relationships with the roots of most terrestrial plant species and represent one of the most abundant organisms in the world [15]. In addition, AMF connects plants with the network of soil fungal hyphae, which are structures that are highly specialized in the absorption of nutrients and water from the soil, which, in turn, receive carbohydrates obtained from the plant through photosynthesis [16]. Several positive effects, such as a decrease in harmful effects by varying the profiles of phenolic compounds in olive plants and improvement of phenolic and antioxidant composition in wheat grains, among others, have been obtained using AMF inoculation of plants under water stress [17,18,19].

In previous studies by our research group using diverse potato genotypes inoculated with the AMF *Claroideoglomus claroideum* and *Claroideoglomus lamellosum*, beneficial responses were identified in the AMF-induced antioxidant activity and phenolic compounds in both potato leaves and tubers [20,21]. Although positive responses were observed in these studies, the effects of AMF inoculation on potato plants under water stress conditions remain unclear. Based on the above, we hypothesized that inoculation with AMF produces changes in the levels of antioxidant compounds and antioxidant activity, positively influencing yield in potato tubers growing under water stress. Therefore, the objective of this study is to investigate the impact of inoculation with AMF on the levels of antioxidant compounds, mainly hydroxycinnamic acids and total phenol concentrations, and antioxidant activity by different chemical methods commonly used in foods, and how this positively influences the yield of potato tubers under water stress conditions.

## 2. Result

### 2.1. Spores, Extraradical Hyphae, and AMF Colonization

As expected, the different AMF inoculants, alone or in combination, had different patterns in terms of the density of fungal propagules; however, the stress and the tested interactions also had a strong influence on the above experimental variables (Table 1). Regarding the AMF spores (Figure 1A), different trends were observed among the treatments. In the CC treatment, an increase in the spore density under water stress conditions was observed, whereas, in the HMC26 treatment, the spore number depended on the stress level, reaching the highest values (675 spores in 100 g of substrate) at the highest water stress level. Similar effects to those of the HMC26 strain were detected with the use of MIX inoculant. The assessment of the extraradical AMF hyphae (Figure 1B) revealed that the CC treatment increased only under the most severe water stress conditions (S2). Meanwhile, the HMC26 and MIX treatments showed extraradical AMF hyphae densities similar to that observed for the spore density. Moreover, the HMC26 treatment resulted in the highest percentages of colonization, with 78.9% and 77.8% under the 0 and S2 conditions, respectively (Figure 1C), similar to the trend observed for spores and extraradical hyphae, with a 62.0% decrease under the S1 conditions compared with the level 0 conditions, while in S2, no significant differences were detected compared with the control. In the CC treatment, under the S1 stress level, there was a 38.4% decrease in colonization, while there were no significant differences under the S2 stress level compared to the control without stress. In contrast, with the MIX treatment, no significant differences between 0 and S1 were observed, while under the S2 conditions, a decrease in root colonization of 25.0% compared to the control was recorded.

### 2.2. Biomass Production

The tuber number (Figure 2A) showed a decrease under the S2 stress level, with this decrease being significant with the inoculation of HMC26 and MIX compared with the uninoculated control. The biomass of the tubers (Figure 2B) was higher mainly in treatments without stress, with the highest values measured in the MIX inoculated plants without water stress, and with a significant decrease in the S1 and S2 plants compared with the plants in the treatment without stress. In the WM, CC, and HMC26 treatments, there was a decrease under both levels of stress, which was significant only under the S2 conditions compared to the control without stress. In general, the level of water stress had more influence on biomass production than the inoculation with AMF (Table 1).

### 2.3. Identification and Quantification of Phenolic Compounds in Tubers

In the potato tubers, only two phenolic compounds were detected, corresponding to isomers of caffeoylquinic acid, which were identified as 5-caffeoylquinic acid (HCAD 1) and caffeoylquinic acid isomer (HCAD 2). Both presented a pseudomolecular ion m/z of 353 uma and a product ion *m*/*z* of 191 uma, with a neutral loss of 162 uma, corresponding to caffeic acid. The two identified compounds were quantified using chlorogenic acids as standards (Table 2 and Table 3). The 5-caffeoylquinic acid concentrations ranged between 1.79 and 21.53 mg kg^−1^, with the MIX without water stress treatment showing the highest concentration. On the other hand, the caffeoylquinic acid isomer concentrations were found to be between 0.80 and 7.40 mg kg^−1^, with the highest concentration detected in the CC-S2 treatment (Table 3). In general, 5-caffeoylquinic acid had higher concentrations than the caffeoylquinic acid isomer in almost all treatments. Regarding the sum of the two identified compounds (HCAD TOT) (Figure 3A), in the WM treatment, a decrease in levels was observed under water stress, with a reduction of 66.1% under the S1 conditions and 68.2% under the S2 conditions compared with the control well-watered. In contrast, in the CC treatment, there was a trend to increase in the presence of water stress, reaching 62.1% for S1 and 129.3% for S2 compared with the control without stress. A similar effect was observed in the HMC26 treatment, where increments of 201.5% for S1 and 72.7% for S2, compared to the control, were reached. In the MIX treatment, HCAD TOT concentrations decreased under water stress, reaching a 57.4% reduction in S1 and 83.7% in S2, compared with the well-watered treatment. In general, for the HCAD TOT, there was a trend to increase with the individual use of AMF and water stress, while in the WM and MIX treatments, there was a trend to decrease.

Regarding total phenol concentrations (Figure 3B), an average of 826 mg kg^−1^ was obtained, without significant differences between treatments. However, a tendency to increase was observed in the uninoculated treatment, as well as under water stress, and with the inoculation of CC and HMC26 strains. This trend was less evident in the MIX treatment, with the concentrations decreased in stress level S2.

### 2.4. Antioxidant Activity

The antioxidant activity determined using the TEAC method (Figure 4A) showed a consistent increase under the application of water stress irrespective of the inoculation treatments. Regarding the CUPRAC method (Figure 4B), the highest activities were detected in the WM treatment; however, an increase was observed when the CC and HMC26 strains were inoculated in treatments subjected to water stress. Regarding the antioxidant activity with the DPPH method, the obtained values were below the detection limit. Regarding the antioxidant activity determined using the ORAC method (Figure 4C), values between 500 and 1721 µmol 100 g^−1^ were registered. The highest activity was found in the CC-0 treatment, showing a significant decrease under both water stress levels. In contrast, with the use of HMC26 as an inoculant, there was a significant increase of 133% in the stress level S2 compared with the control. In the case of MIX treatment, an increase was observed under both water stress levels. In the case of WM-S1 treatment, there was an increase of 74% in their ORAC antioxidant activity, compared with the control.

### 2.5. Multivariate Analysis

A principal component analysis (PCA) was performed considering all the evaluated experimental response variables (Figure 5). An association was observed between the HMC26-0 and MIX-0 treatments and the mycorrhizal traits, as well as with the biomass of the tubers, which can be attributed to these treatments being cultivated without water stress. Moreover, an association between the antioxidant methods ORAC, TEAC, and total phenols (measured according to the Folin-Ciocalteu method) was observed. On the other hand, there was a slight association between the CUPRAC method and the HCAD2 compound.

## 3. Discussion

In a previous related study, the VR808 genotype inoculated with the HMC26 strain achieved a root colonization rate of 20.7%, while inoculation with CC produced only 3.3% root colonization, with both under normal irrigation conditions [20]. The results here suggest that the AMF strain HMC26 is most effective at colonizing plant roots under water restrictions, which may be associated with its origin in the rhizosphere of plants naturally growing in the hyper-arid conditions of the Atacama Desert (Northern Chile). In contrast, other studies have reported that under water shortage, root colonization decreases by approximately 30% compared with the treatment without water stress in soybean plants [22]. A similar trend has been observed in orange seedlings, where colonization with the AMF *Funneliformis mosseae* decreased by 34.4% under water stress compared with treatments with normal irrigation [18]. Whereas, in our work, we detected a significant decrease in the S1 stress level in the individual treatments with mycorrhizae, the treatment with HMC26 presented a higher percentage of colonization and this percentage was maintained with the highest stress level similar to that seen in the other reports.

Regarding biomass production in a related study, Alarcón et al. [20] reported that the VR808 genotype showed a significant increase in tuber number and biomass with the inoculation of the CC and HMC26 strains without water stress. In other studies, compared with non-inoculated treatments, inoculation with *Rhizophagus irregularis* increased potato yield [23]. The same beneficial effect has been reported in wheat, with an increase in yield being associated with inoculation with AMF [24], as well as in rice, in which there has been a positive response in grain growth and yield [25]. Drought stress causes a reduction in leaf water potential and chlorophyll content that reduces the physiological traits of the plant, affecting growth and leaf morphological traits [26]. AMF inoculation can increase biomass accumulation by improving the concentration of macro- and micronutrients, increasing the products of photosynthesis, and concomitantly improving the uptake of inorganic nutrients, especially in nutrient-poor soils [27]. Regarding the effect of water stress, it has been reported that yield is differently affected depending on the potato genotype. For example, for some genotypes, the number of tubers, weight, and yield are not affected, as is the case of the Monalisa cultivar, while in other genotypes, such as Agata and Kennebec cultivars, a higher number of tubers are produced, but with a smaller size according to the water stress increases [9]. The growth period where the stress occurs also affects the yield to a different extent, either producing a decrease in yield or affecting the number of total tubers or their biomass [28]. Similar effects have been reported in wheat crops, where the occurrence of water stress events during the pollen development stage is related to a marked reduction in the number of grains, and if the water starvation event occurs in anthesis, the grain size is reduced [29]. It is generally recognized that the most sensitive periods in potatoes to water stress are the vegetative period and the tuberization stage. When stress occurs in an early stage of sprouting, smaller plants appear, which limits photosynthesis throughout the growth of the plant, restricting resources for tuber production and ultimately producing fewer or smaller tubers [30]. On the other hand, we observed that, despite mycorrhizal inoculation, water stress negatively affected tuber numbers and tuber biomass. Our results highlight that there was a higher colonization in HMC26 0 and S2. However, no increase in biomass was detected in these treatments, moreover, at the S2 level of stress, there was a significant decrease in biomass, independent of the mycorrhizal treatment.

On the other hand, potatoes contain a wide variety of phenolic compounds, such as hydroxycinnamic and hydroxybenzoic acids, flavonols, and flavones, both in free and bound forms [31]. In fleshed-colored genotypes, anthocyanins, such as glycosylated derivatives derived from petunidin in potato tubers, have also been reported [20]. The results reported here are in agreement with those of previous studies on tubers of the VR808 genotype, with two hydroxycinnamic acids (HCADs) derived from chlorogenic acid also being detected, corresponding to 5-caffeoylquinic acid and caffeoylquinic acid isomers [20].

Our previous studies reported HCAD concentrations on the same order of magnitude. For instance, Alarcón et al. [20] reported concentrations of total HCADs between 3.4 and 27.9 mg kg^−1^, reaching the highest value in the control treatment (without AMF) and in the treatment inoculated with the HMC26 strain. However, in such experiments, water stress was not considered. Moreover, it has previously been reported that *C. claroideum* increased the phenolic compounds in the leaves of wheat plants growing under water stress conditions [32]. In other plant species, such as olive, trifoliate orange, and wheat grains, a positive effect has also been reported with the use of AMF, with phenolic compound concentrations increased underwater starvation compared with the treatments without AMF inoculation [17,18,19]. Total phenol concentrations were higher here than those reported by Alarcón et al. [20]. Under abiotic stress conditions, especially environmental stress (e.g., UV radiation), the plant produces ROS and in response; antioxidants, such as phenolic compounds (flavonoids, tannins, hydroxycinnamate esters, and lignin), that play the role of protecting the plant by detoxifying ROS, protect the plant from drought and help to the stabilization of proline and amino acid [33,34]. Water stress can modulate the biosynthetic phenylpropanoid pathway since it regulates genes that encode enzymes, stimulating the biosynthesis of phenolic compounds, which also regulates the biosynthetic pathways of phenolic acids and flavonoids, ultimately leading to the accumulation of these kinds of compounds in different plant species [35].

The concentrations of single compounds quantified using HPLC-DAD were lower and had different trends than those reported using the Folin-Ciocalteu method. A similar behavior was reported in the skin and pulp of *Solanum tuberosum* subsp. Andigena, with the phenolic acids estimated using HPLC-DAD, which presented lower values than those obtained by the Folin-Ciocalteu method [36], may be due to nonspecific reaction in the Folin-Ciocalteu method with compounds such as ascorbic acid, proteins, and some inorganic substances, among others [37]. Nevertheless, the Folin-Ciocalteu assay continues to represent an efficient and economical method for the estimation of total phenols [36]. Therefore, the trend of increasing HCAD seen against both stress levels under both individual mycorrhizal treatments (CC and HMC26) may be due to stimulation by the mycorrhiza as a defense mechanism against water stress that was not seen in the non-mycorrhizal treatment.

Regarding antioxidant activity, a decrease in the TEAC and CUPRAC values in leaves of the VR808 genotype has been previously reported under water stress [38]. Detailing, Fritz et al. [21] reported an increase in the antioxidant activities by TEAC and CUPRAC methods in leaves of the VR808 genotype when CC and HMC26 strains were inoculated. In tubers, a decrease in antioxidant activity has been also reported under similar conditions, but without water stress [20]. Moreover, Tereucán et al. [32] reported an increase in enzymatic and nonenzymatic antioxidant activities in *T. aestivum* leaves under inoculation with *F. mosseae* and *C. claroideum* in response to water stress. The ORAC method is widely used to determine antioxidants in foods and other matrices [39] because it considers the latent phase of antioxidant activities, which is beneficial when measuring foods with slow- and fast-acting antioxidants [37]. For this reason, their evaluation of potato tubers as functional foods is important. In previous studies, ORAC values were reported with values in a range between 24.7 and 346.5 mol TE g^−1^ dry weight (DW) in purple and yellow-fleshed potato cultivars, with values varying considerably depending on the genotype and the environment in which they were grown [40]. In the potato cultivars Blue Bell (yellow skin and flesh with blue spots) and Melody (yellow skin and flesh), the ORAC values were reported to be between 0.233 and 0.222 µmol TE g^−1^ DW [40]. Compared with our work, we observed that for the TEAC method, there was an increase in antioxidant activity as a response to water stress independent of mycorrhizal treatment, while for CUPRAC, there was an increase due to stress in the CC and HMC26 mycorrhizal treatments. Based on these antecedents, to determine the influence of mycorrhizae under water stress, the antioxidant activity should be analyzed using the CUPRAC method.

On the principal component analysis, the association observed between ORAC, TEAC, and total phenols may be due to the method of Folin-Ciocalteu presents high linear correlations with tests based on electron transfer (ET-based) reactions, such as TEAC and DPPH [37]. These traits were also associated with the WM-S1 and WM-S2 treatments, which evidenced the occurrence of water stress conditions specifically in the non-AMF inoculated treatments, where the greater presence of ROS causes metabolic responses leading to the production of a greater amount of antioxidant compounds such as phenolic compounds [41]. This observation is of great importance because it can represent a noticeable and efficient protective role of AMF for coping with drought stress.

It has been reported that berry fruits that present a high amount of HCAD also present high CUPRAC values [42]. In addition, the CUPRAC method has been shown to have advantages compared with other ET-based methods in which hydroxycinnamic acids are also measured [43]. Here, the HMC26-S1 treatment was also mainly associated with HCADTOT, and antioxidant activity was determined with the CUPRAC method. In general, the results presented in this work demonstrate a clear positive effect, mainly of the individual use of AMF, on the antioxidant behavior and protection to cope with drought stress. The differences observed between the different AMF isolates and the magnitude of the effect, especially in combination, could represent other ecological relationships, which supports the need for more studies in the search for effective and sustainable options to increase plant production under the currently prevalent water scarcity scenarios.

A different behavior was observed with the two strains inoculation, highlighting that, regarding yield, total biomass decreased when hydric stress increased, independent of the AMF strain (individual strain or MIX). On the other hand, only CC strain tends to increase the total hydroxycinnamic acid concentrations, whereas MIX or WM treatments tend to decrease this parameter. In the case of total phenols, no significant effects were observed. Finally, the antioxidant activity using the TEAC method increased only in S2 under HMC26 strain and in WM treatment, whereas with CC and MIX a tended increase was detected, whereas using the CUPRAC method, this activity only increased with CC strain and higher water stress (S2). 

Finally, there is not a clear beneficial effect between water stress and AMF inoculation in yield, however important effects in the antioxidant activity were detected with both individual strains.

## 4. Materials and Methods

### 4.1. Reagents

The reagents ABTS (2,2-azino-bis-(3-ethylbenzothiazoline-6-sulfonic acid)), DPPH (2,2-diphenyl-1-picrylhydrazyl), Trolox (6-hydroxy-2,5,7,8-tetramethylchromane-2-carboxylic acid), neocuproine (≥98%), gallic acid, sodium carbonate, chlorogenic acid (>95% purity), fluorescein sodium salt and AAPH (2,2′-azobis(2-amidinopropane)dihydrochloride) were obtained from Sigma–Aldrich (Steinheim, Germany). The neochlorogenic (≥95% purity) and cryptochlorogenic (≥98% purity) acids were purchased from Phytolab (Vestenbergsgreuth, Germany). The reagents water, methanol, ammonia, acetonitrile (HPLC grade), formic acid (p.a. grade), Folin–Ciocalteu reagent, copper (II) chloride (p.a. grade), potassium phosphate monobasic and potassium phosphate dibasic were obtained from Merck (Darmstadt, Germany).

### 4.2. Biological Material and Experimental Design

Potato seeds of the VR808 genotype, characterized by yellow pulp and skin, were provided by Papas Arcoiris Ltd.a. (Puerto Varas, Chile). The *S. tuberosum* plants were grown according to seasonality in the greenhouse of the Departamento de Ciencias Químicas y Recursos Naturales, Universidad de La Frontera (Temuco, Chile) (38°44′50.2″ S 72°36′54.2″ W), where they were subjected to 50% artificial shade by placing a plastic mesh over the greenhouse, maintaining a light/dark photoperiod of 16/8 h with supplementary LED light as needed and day-night temperatures of 25/18 °C, in 11 L pots. A loamy soil diluted with sand (2:1 soil/sand, *v*/*v*) was used as substrates, and autoclave-sterilized (121 °C for 20 min on 3 consecutive days). The substrate properties were pH 6.7 (in water, 2:5 *w*/*v*); 5.0% organic matter; nutrient concentrations of N 21, P 6, and K 29 (mg kg^−1^); and electrical conductivity (EC) of 0.09 μS cm^−1^ (1:5, *w*/*v*). Two strains of AMF were used as inoculants: *Claroideoglomus claroideum* (CC) and *C. lamellosum* (HMC26) (accession number MN263071). CC was obtained from crops in the Araucanía Region (Chile) and was associated with the rhizosphere of wheat plants, and HMC26 was obtained from the rhizosphere of *Baccharis scandens* (Ruiz y Pav.) (Asteraceae) plants in the Camiña Valley, Atacama Desert (Tarapacá Region, northern Chile). In both cases, the reproduction of AMF propagules was carried out in trap pots using *Bidens pilosa* as a host plant growing for six months. Both substrates were sieved using sieves of 500 and 53 μm, including spores (around 700 spores per g) and abundant mycelium, and this sieved substrate was used as AMF inoculum. The following treatments were used: (i) CC, (ii) HMC26, (iii) MIX (CC + HMC26), and (iv) a control (WM: without AMF). For inoculation, the methodology described by Valdebenito et al. [38] was used, with 5 g of the AMF inoculum added to the substrate below the tuber in each pot (approximately 700 spores per gram). For crop management, the AMF inoculum was added to each of the treatments, including the control and the commercial fungicide REFLEXTRA^®^ Syngenta Ltd.a. (Grangemouth, UK) according to the manufacturer’s instructions. After seeding, the plants were treated with normal irrigation until the beginning of tuberization, with water stress applied under three conditions: 100% (0), 70% (S1), and 40% (S2) humidity levels relative to the water holding capacity in the soil. The three irrigation levels were maintained throughout the experiment by weighing the pots every 2 days. Each treatment was performed in triplicate plus a control without AMF or water stress (*n* = 36). The crop was fertilized with nitrogen, phosphorus, and potassium. In detail, 0.45 g N/plant of urea-N, 0.28 g P_2_0_5_/plant, and 0.35 g K_2_0/plant, were used as basal application. The experiment was finished when the tubers were harvested, which occurred approximately 180 days after planting. Once the tuberization process was complete, the number of tubers and biomass production were evaluated in each pot according to the method reported by Alarcón et al. [20].

### 4.3. Spores, Extraradical Hyphae, and AMF Colonization

The quantification of AMF spores was carried out according to the method described by Sieverding [44] with minor modifications consisting of the use of 20 g of soil sample, the use of a previous step using tap water to eliminate dead spores and gross organic materials, and the use of only two sieves of 0.5 mm and 36 nm to separate the spores. The spores were quantified using a stereoscopic magnifying glass with a magnification of 40–90× in a Doncaster dish. The quantification of the AMF mycelia was carried out as described by Borie and Rubio [45]. The intersections between the extraradical mycelium and the nitrocellulose filter gridline were quantified under a stereoscopic microscope (100–400×), and the density of the AMF mycelia was calculated using Newman’s formula [46], AMF colonization was analyzed by staining several root pieces of approximately 1 cm in length with a 0.05% trypan blue solution in lactic acid, and three spots per root were quantified using a grid [47].

### 4.4. Extraction of Antioxidant Compounds

For the extraction of phenolic compounds, the methodology described by Alarcón et al. [20] with modifications was used. Briefly, 1 g of ground raw potato was mashed and homogenized with 3 mL of the extraction solvent methanol:formic acid 97:3 *v*/*v*. Then, ultrasound was applied for 60 s at 40% amplitude with an ultrasonic processor at 130 W (Sonics and Materials, Newtown, CT, USA) and the samples were mixed in an orbital shaker for 10 min at 200× *g*. Finally, the extract was centrifuged (Lab Companion, Seoul, Republic of Korea) for 15 min at 4000× *g* at 4 °C. The extraction process was performed twice. Once the extract was obtained, it was filtered through 0.45 μm filters and stored at −20 °C in the dark until analysis.

### 4.5. Identification and Quantification of Phenolic Compounds

The analysis was performed as described by Alarcón et al. [20] using a high-performance liquid chromatography–diode array detection (HPLC-DAD) system (Shimadzu, Tokyo, Japan) equipped with an LC-20AT quaternary pump, a DGU-20A5R degassing unit, a CTO-20A oven and a diode array detector (SPD-M20A). Identity assignments were performed using an HPLC-DAD system coupled to an MDS Sciex system QTrap3200 liquid chromatography–tandem mass spectrometry instrument (Applied Biosystems, Foster City, CA, USA) and confirmed by comparison with the retention time when utilising commercial standards. A C_18_ column (250 × 4.6 mm, 5 μm) (Kromasil, Supelco, Bellefonte, PA, USA), a C_18_ precolumn (22 × 3.9 mm, 4 μm) (Novapak, Waters, Milford, MA, USA) and an oven temperature of 40 °C were used. Two mobile phases were used: mobile phase A was composed of water:acetonitrile:formic acid (87/3/10 *v*/*v*/*v*), while mobile phase B was composed of water:acetonitrile:formic acid (40/50/10 *v*/*v*/*v*). Hydroxycinnamic acid (HCAD) detection was performed at 320 nm by external calibration using chlorogenic acid as a standard.

Total phenols were determined using the method described by Parada et al. [48] in 96-well microplates using a microplate reader SYNERGY HTX (BioTek Instruments, Winooski, VT, USA) to obtain the absorbance at 750 nm. Gallic acid was used as a standard.

### 4.6. Determination of Antioxidant Activity

The antioxidant activity was determined using the copper-reducing antioxidant capacity (CUPRAC), Trolox equivalent antioxidant capacity (TEAC), and 2,2-diphenyl-1-picrylhydrazyl radical scavenging (DPPH) methods, as described by Parada et al. [48] The antioxidant activity was also determined by the total antioxidant capacity by the oxygen radical absorbance capacity (ORAC) method as described by Ou et al. [49] Measurements were performed in a microplate reader SYNERGY HTX (BioTek Instruments, Winooski, VT, USA). Trolox was used as the standard for all methods, and the results were expressed as Trolox equivalents (TE).

### 4.7. Statistical Analysis

The datasets were analyzed using two-way analysis of variance (ANOVA) with AMF inoculation and water stress level as well as the interaction between the two as the main variation sources. The ANOVA was performed after checking the normality and homoscedasticity of the data. The means were compared post hoc using Tukey’s multiple range test, and a significance level of *p* < 0.05 was established as significant for all cases. Additionally, a factorial analysis with the extraction of principal components was performed, with the fixed variables considered being the different AMF isolates and the respective uninoculated control and the two levels of stress plus the nonstressed control. The software used for the analysis was IBM SPSS Statistics v. 23 (IBM Corp., New York, NY, USA).

## 5. Conclusions

Here, inoculation with arbuscular mycorrhizal fungi (AMF), especially monospecific inoculants, showed clear improvements mainly in the antioxidant status of potato tubers produced under water stress conditions. However, the effect was dependent on the AMF strain and the severity of the water stress applied. We found that the use of *Claroideoglomus lamellosum* had a better effect on the experimental variables analyzed than *C. claroideum* and the mixture of both, suggesting that the prevalent conditions of the environment where the AMF is obtained are relevant in the search for effective biotechnological tools for improving plant production. Moreover, despite the inoculation factor, the yield was strongly affected by the different conditions of water stress. This underscores the need to carry out more studies that include other varieties of *S. tuberosum* and isolates of AMF to confirm that the directed use of AMF, alone or in consortium, has beneficial effects on potato crops affected by water stress.

## Figures and Tables

**Figure 1 plants-12-04171-f001:**
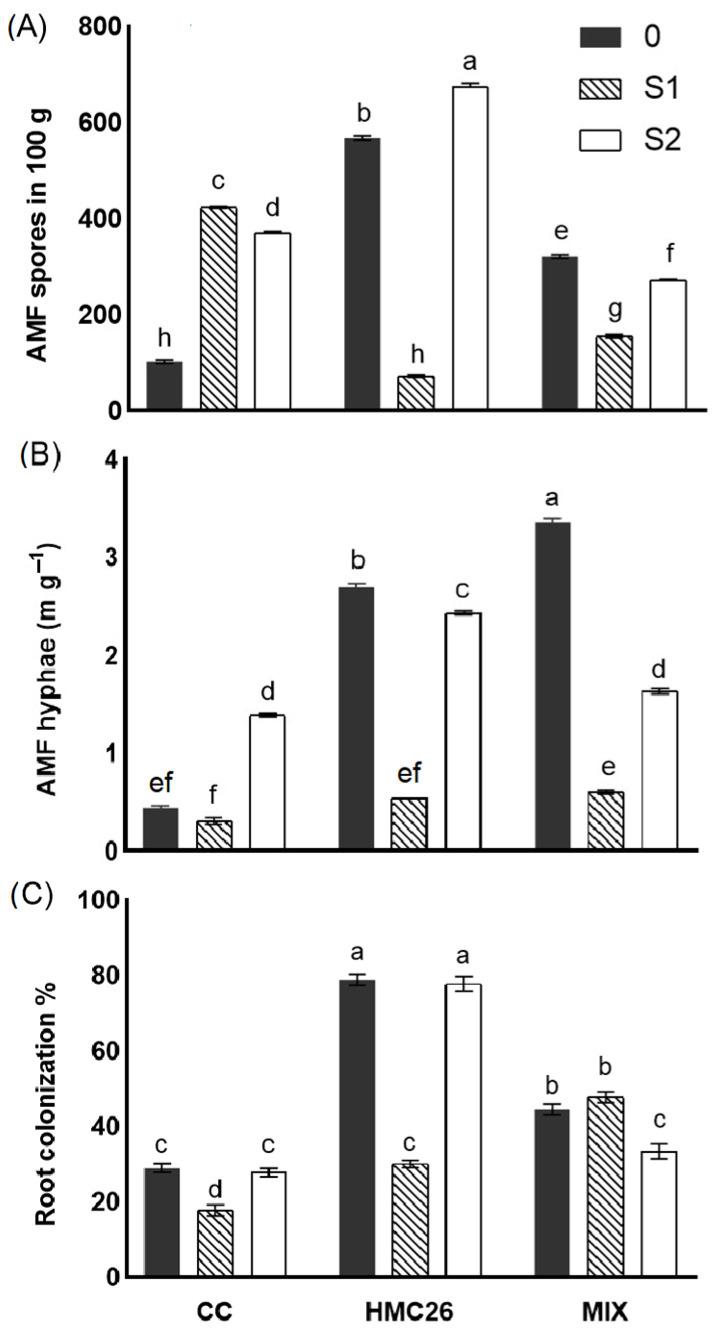
Quantification of arbuscular mycorrhizal fungi (AMF) characteristics in *Solanum tuberosum* plants growing under normal irrigation and two conditions of water stress: (**A**) AMF spores in the substrate; (**B**) extra radical AMF hyphae; (**C**) AMF root colonization. WM: without mycorrhiza; CC: *Claroideoglomus claroideum*; HMC26: *Claroideoglomus lamellosum*; MIX: CC + HMC26 and 0:100%, S1: 70%, S2: 40% humidity levels in soil. Different letters indicate significant differences according to Tukey’s multiple range test (*p* < 0.05).

**Figure 2 plants-12-04171-f002:**
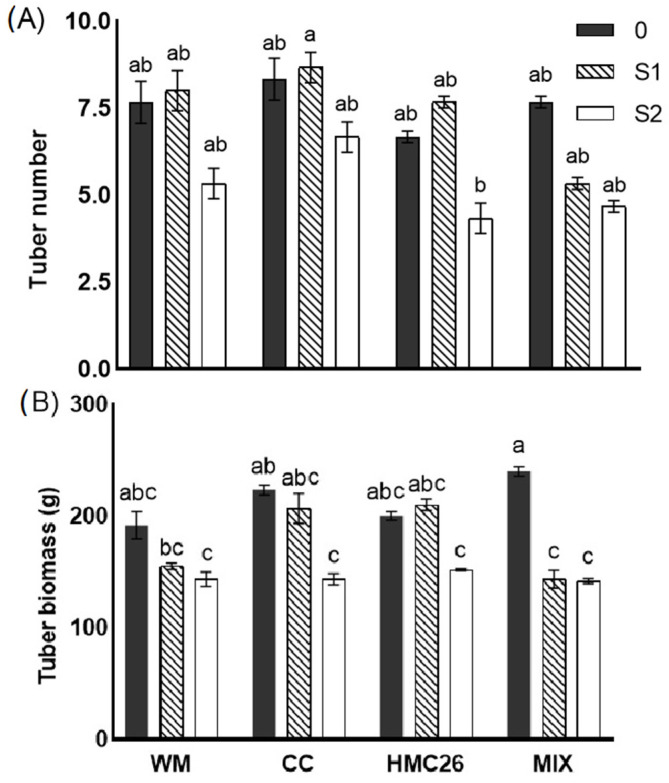
Yield parameters of *Solanum tuberosum* tubers growing under normal irrigation and two conditions of water stress: (**A**) number of tubers; (**B**) tuber biomass. WM: without mycorrhiza; CC: *Claroideoglomus claroideum*; HMC26: *Claroideoglomus lamellosum*; MIX: CC + HMC26 and 0:100%, S1: 70%, S2: 40% humidity levels in soil. Different letters indicate significant differences according to Tukey’s multiple range test (*p* < 0.05).

**Figure 3 plants-12-04171-f003:**
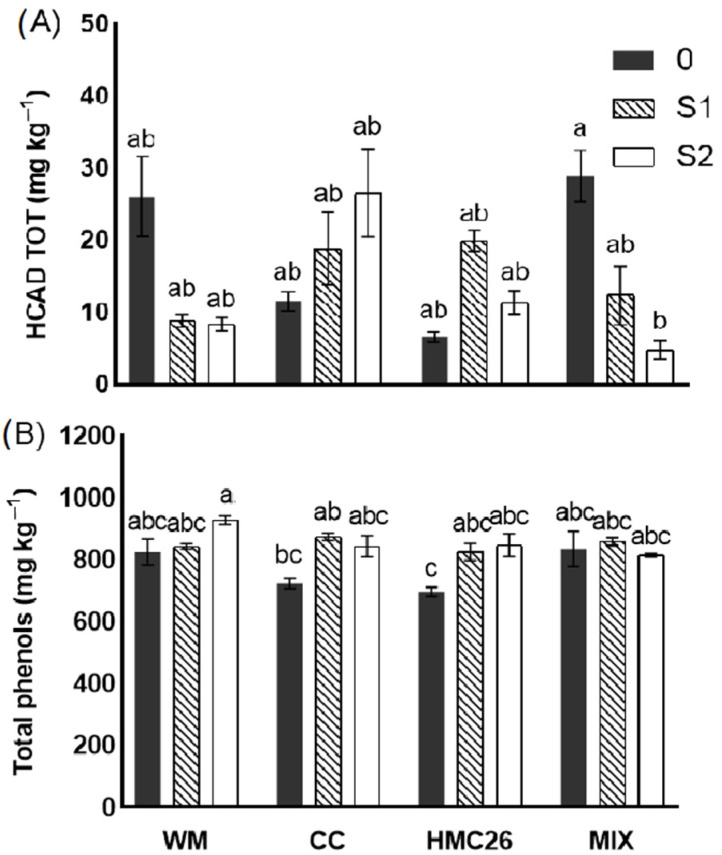
Phenolic compound concentrations (fresh weight) in tubers of *Solanum tuberosum* growing under normal irrigation and two conditions of water stress: (**A**) total hydroxycinnamic acids (HCAD TOT); (**B**) total phenols determined using the Folin-Ciocalteu method. WM: without mycorrhiza; CC: *Claroideoglomus claroideum*; HMC26: *Claroideoglomus lamellosum*; MIX: CC + HMC26 and 0:100%, S1: 70%, S2: 40% humidity levels in soil. Different letters indicate significant differences according to Tukey’s multiple range test (*p* < 0.05).

**Figure 4 plants-12-04171-f004:**
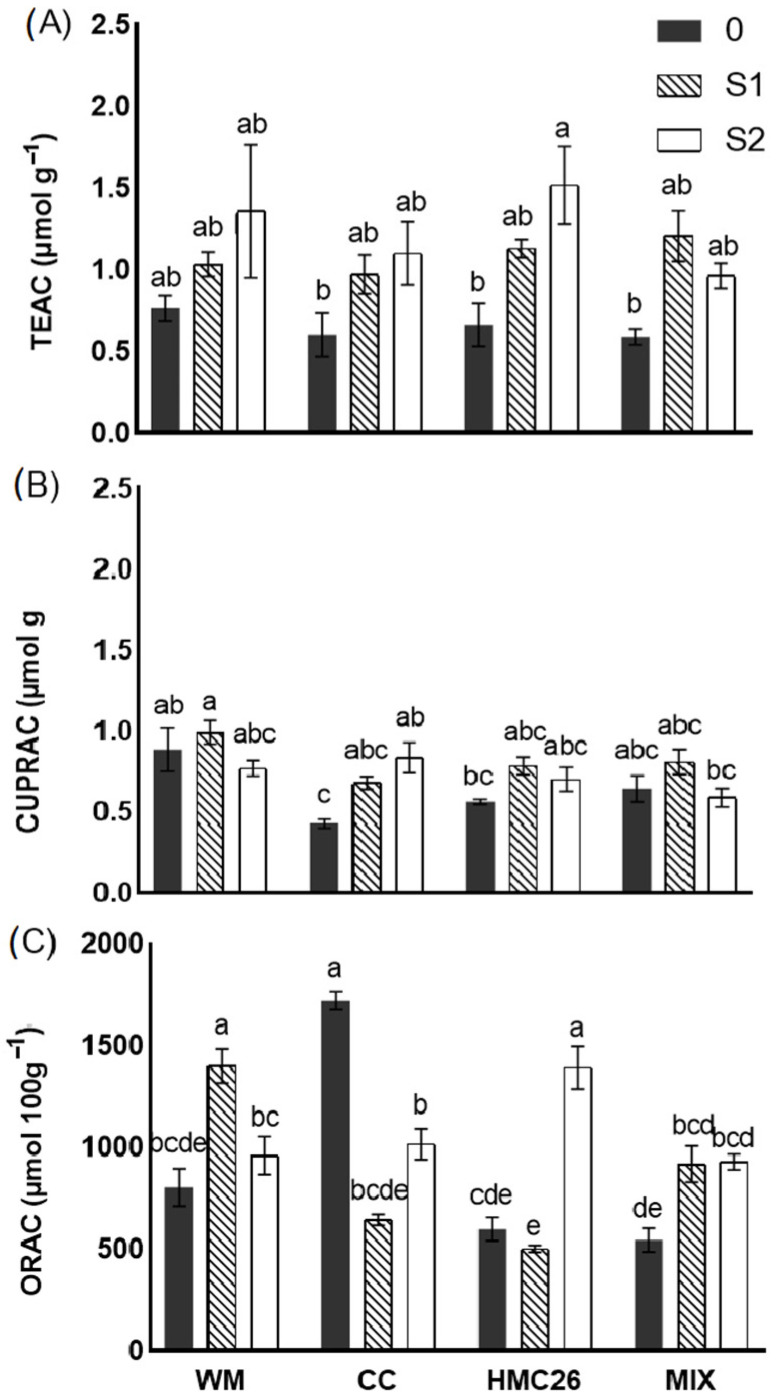
Antioxidant activity (AA) (fresh weight) in tubers of *Solanum tuberosum* growing under normal irrigation and two conditions of water stress: (**A**) AA determined using the Trolox equivalent antioxidant capacity (TEAC) method; (**B**) AA determined using the copper-reducing antioxidant capacity (CUPRAC) method; (**C**) AA determined using Total Antioxidant Capacity using Oxygen Radical Absorbance Capacity (ORAC) method. WM: without mycorrhiza; CC: *Claroideoglomus claroideum*; HMC26: *Claroideoglomus lamellosum*; MIX: CC + HMC26 and 0:100%, S1: 70%, S2: 40% humidity levels in soil. Different letters indicate significant differences according to Tukey’s multiple range test (*p* < 0.05).

**Figure 5 plants-12-04171-f005:**
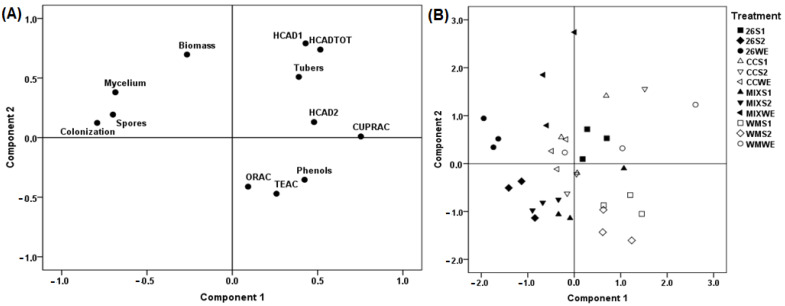
(**A**) Principal component (PC) scores for the experimental variables determined in tubers of *Solanum tuberosum* plants inoculated or not with different strains of arbuscular mycorrhizal fungi (AMF) and that grew under different irrigation conditions. The percentage values in brackets indicate the variation explained by each PC. (**B**) The graph shows the distribution of experimental individuals according to PC and grouped according to treatment. HCADTOT: sum of HCAD1 and HCAD2; HCAD1: 5-caffeoylquinic acid; HCAD2: caffeoylquinic acid isomer; Tubers: number of tubers; Biomass: Biomass of tuber; Phenols: total phenols determined by the Folin-Ciocalteu method; Spores: AMF spores in the substrate; Mycelium: extraradical AMF hyphae; Colonization: AMF root colonization. WM: without mycorrhiza; CC: *Claroideoglomus claroideum*; 26: *Claroideoglomus lamellosum*; MIX: CC + HMC26 and 0:100%, S1: 70%, S2: 40% humidity levels in soil.

**Table 1 plants-12-04171-t001:** *F*-values and probabilities of significance for the main effects and factor interaction for the variables were measured and analyzed using two-way ANOVA.

Source of Variation	HCAD1	HCAD2	HCAD TOT	Phenols	TEAC	CUPRAC	ORAC	Tubers	Biomass	Mycelium	Spores	Colonization
AMF	0.94 ^ns^	3.49 *	0.49 ^ns^	1.27 ^ns^	1.19 ^ns^	2.48 ^ns^	5.39 **	3.62 *	2.23 ^ns^	1041 ***	2649 ***	316 ***
Water stress	1.25 ^ns^	0.26 ^ns^	0.70 ^ns^	3.82 *	13.95 ***	2.50 ^ns^	3.16 ^ns^	10.31 **	25.57 ***	718 ***	732 ***	33.3 ***
AMF x Stress	2.78 *	2.39 ^ns^	2.46 ^ns^	0.93 ^ns^	0.93 ^ns^	1.13 ^ns^	14.2 ***	0.96 ^ns^	2.76 *	256 ***	990 ***	33.1 ***

Significance conventions: * *p* < 0.05; ** *p* < 0.01; *** *p* < 0.001; ns: no significant differences; AMF: arbuscular mycorrhizal fungi; HCAD 1: 5-caffeoylquinic acid; HCAD 2: caffeoylquinic acid isomer; T HCAD: total of hydroxycinnamic acids.

**Table 2 plants-12-04171-t002:** Analytical parameters for determinations of phenolic compounds by HPLC-DAD and antioxidant activities by spectrophotometric methods.

Method	Standard	Equation	R^2^	DL	QL	LR
HCAD TOT	Chlorogenic acid	y = 73284x + 6553.5	1.000	0.042 mg L^−1^	0.140 mg L^−1^	0.140–100 mg L^−1^
FOLIN	Gallic acid	y = 0.0006x + 0.0008	0.996	3.600 mg L^−1^	12.001 mg L^−1^	12.001–500 mg L^−1^
TEAC	Trolox	y = 0.4094x + 0.013	0.991	0.024 mmol L^−1^	0.078 mmol L^−1^	0.045–0.7 mmol L^−1^
CUPRAC	Trolox	y = 3.2224x + 0.1266	0.996	0.008 mmol L^−1^	0.027 mmol L^−1^	0.027–0.6 mmol L^−1^
DPPH	Trolox	y = 0.7608x + 0.0272	0.990	0.036 mmol L^−1^	0.119 mmol L^−1^	0.119–0.7 mmol L^−1^
ORAC	Trolox	y = 0.38x + 6.7092	0.995	0.870 umol L^−1^	2.899 umol L^−1^	2.899–80 umol L^−1^

HCAD TOT: Total hydroxycinnamic acids, TEAC: Trolox equivalent antioxidant capacity, CUPRAC: Cupric reducing antioxidant capacity, DPPH: Antioxidant activity by the 2,2-diphenyl-1-picrylhydrazyl radical scavenging (DPPH) method, ORAC: Oxygen Radical Absorbance Capacity, DL: Detection limit, QL: Quantification limit, LR: Linear range.

**Table 3 plants-12-04171-t003:** Individual phenolic compounds concentration (mg kg^−1^ fresh weight) by HPLC-DAD in *Solanum tuberosum* tubers. Different letters indicate significant differences according to Tukey’s multiple range test (*p* < 0.05).

Treatment	5-Caffeoylquinic Acid (mg kg^−1^)	Caffeoylquinic Acid Isomer (mg kg^−1^)
WM 0	21.06 ± 15.94 ^ab^	5.00 ± 3.49 ^ab^
CC 0	10.20 ± 5.32 ^abc^	1.36 ± 0.66 ^b^
HMC26 0	4.73 ± 2.22 ^abc^	1.83 ± 0.35 ^ab^
MIX 0	21.53 ± 12.27 ^a^	7.35 ±1.39 ^a^
WM S1	2.40 ± 1.15 ^bc^	6.44 ± 2.08 ^ab^
CC S1	18.02 ± 16.78 ^abc^	0.80 ± 0.74 ^b^
HMC26 S1	18.98 ± 4.81 ^abc^	0.90 ± 0.56 ^b^
MIX S1	5.09 ± 6.66 ^abc^	7.25 ± 7.59 ^a^
WM S2	3.29 ± 1.77 ^abc^	5.02 ± 1.48 ^ab^
CC S2	19.18 ± 17.73 ^abc^	7.40 ± 3.80 ^a^
HMC26 S2	8.13 ± 4.88 ^abc^	3.27 ± 2.12 ^ab^
MIX S2	1.79 ± 1.94 ^c^	2.95 ± 2.60 ^ab^

## Data Availability

The data presented in this study are available on request from the corresponding author.

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
