# Peer review of "Antioxidant Response, Phenolic Compounds and Yield of Solanum tuberosum Tubers Inoculated with Arbuscular Mycorrhizal Fungi and Growing under Water Stress"

_plants, 2023, doi:10.3390/plants12244171_

Round 1

Reviewer 1 Report

Comments and Suggestions for Authors

This study emphasizes the potential role and possible application of AMF in potato cultivation for sustainable crop growth and management. There are certain merits in this work, but substantial improvement is required.

1.      The results mention changes in yield, phenolic profiles, and antioxidant activity, but specific data points or statistical values are not presented in the summary. Providing actual numbers and statistical significance would improve the transparency and allow for a more comprehensive evaluation.

2.      Please improve the sentence in the abstract line 19-21.

3.      Line 50-52 not linked to previous sentence.

4.      Is it true “The optimal temperatures for tuber formation are between 15 and 25 52 °C, and the tuberization signal can be inhibited at higher temperatures” what are the references for that.

5.      The objectives of the study could be more explicitly stated. While the general aim is to evaluate the impact of AMF on potato under water stress, the specific goals and hypotheses are not clearly outlined.

6.      What was the plot size in experimental site?

7.      Please write AMF name in italics and give accession numbers of these strains

8.      Fig 1: WM?

9.      Give comparative photographs of Significant results of treatment with AMF. 

Comments on the Quality of English Language

Please improve the sentences in abstract, introduction and discussion. 

Author Response

REVIEWER 1

This study emphasizes the potential role and possible application of AMF in potato cultivation for sustainable crop growth and management. There are certain merits in this work, but substantial improvement is required.

Dear reviewer, thank you very much for your valuable comments. Here, we give response to all your questions and comments.

  1. The results mention changes in yield, phenolic profiles, and antioxidant activity, but specific data points or statistical values are not presented in the summary. Providing actual numbers and statistical significance would improve the transparency and allow for a more comprehensive evaluation.

R: Thank you for the suggestion. We have extensively changed the abstract, also including specific details that evidence the role of the AMF inoculation on the levels of HCAD and antioxidant activities. Please, see the new version.

  1. Please improve the sentence in the abstract line 19-21.

R: Corrected.

  1. Line 50-52 not linked to previous sentence.

R:  The redaction was corrected.

  1. Is it true “The optimal temperatures for tuber formation are between 15 and 25 52 °C, and the tuberization signal can be inhibited at higher temperatures” what are the references for that.

R: Thank you very much for your comment. The reference was missing and now it was incorporated in text (L54-55). 

  1. The objectives of the study could be more explicitly stated. While the general aim is to evaluate the impact of AMF on potato under water stress, the specific goals and hypotheses are not clearly outlined.

R: Effectively, in our study we indicated the hypothesis and only the main aim. In this new version, more detailed information was added to clarify the aims of the work. Please, see L91-95.

  1. What was the plot size in experimental site?

R: The experiment was designed to be performed in pots of 11 L, which is, according to our previous experience, a size ideal to warrant an adequate potato grow for the genotypes here used. The details were included in the second section in the material and methods chapter. Thank you for your question.

  1. Please write AMF name in italics and give accession numbers of these strains

R:  The inoculum of Claroideoglomus lamellosum was reproduced and identified by Santander et al. (2021), whose sequences were deposited in the NCBI database with the sequence accession number MN263071. The sequences of Claroideoglomus claroideum are currently in the process of being deposited in the NCBI database.

  1. Fig 1: WM? 

R: WM: without mycorrhiza.

The information was included in the original manuscript and now is highlighted in blue (please see L122-123).

  1. Give comparative photographs of Significant results of treatment with AMF.

R: Dear reviewer, thank you very much for your comment. Unfortunately, because the samples were processed last year and the trial was over, we do not have photographs to compare the significant results of the AMF treatments, however, we appreciate your suggestions and will include these photographs in a future trial.

Reviewer 2 Report

Comments and Suggestions for Authors

The manuscript entitled "Antioxidant Response, Phenolic Compounds and Yield of Solanum tuberosum Tubers Inoculated with Arbuscular Mycorrhizal Fungi and Growing Under Water Stress” evaluated yield, phenolic profiles, and antioxidant activity in tubers of the VR808 potato genotype grown at increasing levels of water stress, namely, with 100% (0), 70% (S1), and 40% (S2) soil humidity and inoculated with two arbuscular mycorrhizal fungi (AMF) strains, Claroideoglomus claroideum (CC), Claroideoglomus lamellosum (HMC26) and the MIX (CC+HMC26).

Comments:

The experimental design needs to be explained with more specific details. For example, which kind of AMF inoculation was used? Does 5 gm of AMF include only spores and mycelia? What are the amounts of soil per pot?

The effect of AMF inoculation on potato plants under water stress should be also evaluated in unsterilized soil to assess the applicability of AMF inoculation.

Moreover, the discussion is somewhat careless in form: an example the result that no interaction was found between water stress and AMF inoculation has to be pointed out. It is particularly important for the plant growth parameters that these two factors work dependent/independent from each other. I have the feeling that the authors did not understand the consequences of this result, where an interaction exists or not. Based on the presented results, this seems to be a myth.

Linkage between AMF inoculation percentages and measured plant biomass should be clearly interpreted. In particular, under different water stress levels.

In the discussion section 238-261, authors discussed the effect of AMF inoculation and water stress on plant biomass for previous studies without explaining and concluding the effect of these treatments on their own study.

The same comment was found in the discussion section regarding antioxidant activity, line 295-311.

Different letters which indicate the significant differences in all figures should be revised because many statistical mistakes were found.

Check the scientific name which should be italic. For example, The S. tuberosum plants were…..Line 347 should be italic. Claroideoglomus claroideum (CC) and Claroideoglomus lamellosum (HMC26)….. Line 358-359.

Comments on the Quality of English Language

Minor editing of English language required

Author Response

REVIEWER 2

The manuscript entitled "Antioxidant Response, Phenolic Compounds and Yield of Solanum tuberosum Tubers Inoculated with Arbuscular Mycorrhizal Fungi and Growing Under Water Stress” evaluated yield, phenolic profiles, and antioxidant activity in tubers of the VR808 potato genotype grown at increasing levels of water stress, namely, with 100% (0), 70% (S1), and 40% (S2) soil humidity and inoculated with two arbuscular mycorrhizal fungi (AMF) strains, Claroideoglomus claroideum (CC), Claroideoglomus lamellosum (HMC26) and the MIX (CC+HMC26).

Comments:

  1. The experimental design needs to be explained with more specific details. For example, which kind of AMF inoculation was used? Does 5 gm of AMF include only spores and mycelia? What are the amounts of soil per pot?

R: Dear reviewer, thank you very much for all your comments.

Two AMF isolates were used: Claroideoglomus Claroideum (CC) obtained from agricultural crops in the Araucania Region (Chile) and which was associated with the rhizosphere of wheat plants, and Claroideoglomus lamellosum (HMC26) obtained from the rhizosphere of Baccharis scandens (Ruiz y Pav.) (Asteraceae) plants in the Camiña Valley, Atacama Desert (Tarapacá Region, northern Chile). In both cases, the reproduction of AMF species was carried out in trap pots using Bidens pilosa as a host plant for six months. Both substrates were sieved using sieves of 500 and 53 mm, including spores (around 700 spores per g) and abundant mycelium, and this sieved substrate was used as AMF inoculum (L386-397).

  1. The effect of AMF inoculation on potato plants under water stress should be also evaluated in unsterilized soil to assess the applicability of AMF inoculation.

R: Dear reviewer, thank you very much for your comment. In this work, as a first instance, we only cover the greenhouse trial under semi-controlled conditions; however, in our next studies we want to be able to take this trial under field conditions to effectively evaluate the applicability of inoculation with AMF in Solanum tuberosum crops.

  1. Moreover, the discussion is somewhat careless in form: an example the result that no interaction was found between water stress and AMF inoculation has to be pointed out. It is particularly important for the plant growth parameters that these two factors work dependent/independentfrom each other. I have the feeling that the authors did not understand the consequences of this result, where an interaction exists or not. Based on the presented results, this seems to be a myth.

R: Thank you very much. Effectively, based on the results, we can observe the following: differentiate behavior was observed with the two strains inoculation, highlighting that, regarding yield, total biomass decreased when hydric stress increase, independent of the AMF strain (individual strain or MIX). On the other hand, only CC strain tends to increase the total hydroxycinnamic acid concentrations, whereas MIX o WM treatments tends to decrease this parameter. In the case of total phenols, no significant effects were observed. Finally, the antioxidant activity by TEAC method increased only in S2 under HMC26 strain and also in WM treatment, whereas with CC and MIX a tend to increase was detected, whereas by CUPRAC method, this activity only increases with CC strain and higher water stress (S2). Finally, there are not a clear beneficial effect between water stress and AMF inoculation in yield, despite important effects in the antioxidant activity were detected with both individual strains. This information was incorporated at the end of the discussion section (L354-364).

  1. Linkage between AMF inoculation percentages and measured plant biomass should be clearly interpreted. In particular, under different water stress levels.

R: Thank you very much for your comment. Our results showed that there was a higher colonization in HMC26 0 and S2. However, no increase in biomass was seen in these treatments; moreover, at the S2 level of stress there was a significant decrease in biomass, independent of the mycorrhizal treatment (see paragraphs 1, 2 and 3 in discussion). In fact, correlation analysis showed that only significant r-Pearson values were obtained between mycorrhization with spores and mycelium (0.806 and 0.783, respectively).

  1. In the discussion section 238-261, authors discussed the effect of AMF inoculation and water stress on plant biomass for previous studies without explaining and concluding the effect of these treatments on their own study.

R: Dear reviewer, thank you very much for your comment. A most comprehensive interpretation was incorporated in the manuscript at discussion section. Please, see paragraphs 1, 2 and 3 in discussion.

  1. The same comment was found in the discussion section regarding antioxidant activity, line 295-311.

R: Dear reviewer, thank you very much for your comment. Your comments were incorporated. Please, see changes in L326-332.

  1. Different letters which indicate the significant differences in all figures should be revised because many statistical mistakes were found.

R: Thank you for your appreciation. In fact, this mistake was previously solved, since the letters to identify the subgroups were erroneous. Unfortunately, we included in the previous version submitted to Plants the initial figures with some mistakes. Please, see the letters in the new figures incorporated in the manuscript.

  1. Check the scientific name which should be italic. For example, The S. tuberosum plants were…..Line 347 should be italic. Claroideoglomus claroideum (CC) and Claroideoglomus lamellosum (HMC26)….. Line 358-359.

R: Sorry for the mistake. It was corrected throughout the manuscript.

Reviewer 3 Report

Comments and Suggestions for Authors

General comments

I have read the manuscript:  plants MDPI. Entitle: Antioxidant Response, Phenolic Compounds and Yield of Solanum tuberosum Tubers Inoculated with Arbuscular Mycorrhizal Fungi and Growing Under Water Stress written by Javiera Nahuelcura et. al., for publication of plants MDPI. In this study, author evaluated the inoculation of potato with two arbuscular mycorrhizal fungi (AMF) strains, Claroideoglomus claroideum (CC), Claroideoglomus lamellosum (HMC26) and the MIX (CC+HMC26). Yield, phenolic profiles, and antioxidant activity at increasing levels of water stress. Result showed that the AMF did not show any influence on the number and biomass of the tubers. Inoculation with the individual AMF strains, especially C. lamellosum, resulted in a better response in terms of phenolic compounds and antioxidant activity measured as Oxygen Radical Absorbance Capacity (ORAC) in the potato tubers under water stress.

The overall research is well conducted and very information because this study suggesting specific AMF adaptations to water stress that must be considered when inoculation. In this sense, this manuscript is much more valuable. However, I found a lack of story connection and lack of potential references (some I suggested some below). Overall after I evaluate and request the author for this manuscript as a “MAJOR REVISION”.

Major Suggestions

1) Abstract: Author should improve abstract further by novelty of the result and behalf of lengthening the methodology part. Author should rephrase the result and make it clear by addressing the common message for audiences. This is very important than only lengthening the text. Author should rephrase the abstract and whole manuscript should elucidate how this study is useful for the society?

2) Introduction: Introduction is well presented the research objectives in the Ln 90-93 in the last section which is highly appreciated. However, please connect the research objectives with the research hypothesis. Author should clearly mention the research hypothesis, and this should relate to the research objectives. The hypothesis and research objectives should be very clear because, without appropriate literature, questions, or hypotheses in the introduction section the entire text will be unclear.

3) Concise the text: author should concise the text by removing the unnecessary and less important text. Please include the text related on the research title and its circumstances by cutback unnecessary text.

Some line-to-Line comments

4) Line no. 39 (Introduction): Author should mention increasing the world population and food hunger scenario and related data, which is much appreciated. Drought generally involves to reduction of morphological characteristics specially alter of leaf size and reduction of plant and leaf water contents, xylem sap flow and whole plant hydraulic conductivity because drought destruction and alter the anatomical characteristics refer this article https://doi.org/10.1016/j.foreco.2020.118099 for the further clarify the negative effect under drought in introduction.

5) Line no. 226 (Discussion): Author should address why arises the antioxidant and secondary metabolites under drought? Please see the Ln 217-319 and Refer to these two articles for better clarify (1) https://doi.org/10.1038/s41598-019-55889 (2) https://doi.org/10.1016/j.scitotenv.2021.146466 and mention somewhere in that paragraph “abiotic stress especially environmental stress (e.g. UV radiation) plant produces the ROS when the plant exposed to the stress condition and plant produce antioxidant, flavonoids, and secondary metabolites , play to the role for protecting the plant for detoxifying ROS and protect the plant to protect from drought and help to stabilization of proline and amino acid”.

6) Line no. 109/201 (Results): The legend should clearly describe about each figure. Accordingly, please check all the table and Figure legends throughout the manuscript. Moreover, the letter inside the figures is not clear in most of the figures please increase its font size and its visibility.

7) Line no. 247-251 (Discussion): Author should further interpretation the negative effect of leaf morphological and growth traits and its related causes. Drought stress reduction of leaf water potential, chlorophyll content that reduction the physiological traits (photosynthesis) of the plant. “Generally, reduction of photosynthesis that effect on the growth and leaf morphological traits under drought”. Please refer this article https://doi.org/10.1016/j.scienta.2018.11.021 and elaborate the text more broadly.

8) Line no. 437 (Conclusion): Conclusion should not be repetitive in the abstract or a summary of the results section. Please reduce the number of paragraphs which is generally not suited in the conclusion. Author should concise the text, I would love to read striking points and take-home messages that will linger in the readers’ minds. What is the novelty, how does the study elucidate some questions in this field, and the contributions the paper may offer to the scientific community?

9) Line no. 467 (References): please include more related citation, check their pattern and writing style, spell check, and other grammatical errors. moreover, the author should cut the old and less matching literature and include the latest literature some of them are above.

Good Luck !

Author Response

REVIEWER 3

General comments

I have read the manuscript:  plants MDPI. Entitle: Antioxidant Response, Phenolic Compounds and Yield of Solanum tuberosum Tubers Inoculated with Arbuscular Mycorrhizal Fungi and Growing Under Water Stress written by Javiera Nahuelcura et. al., for publication of plants MDPI. In this study, author evaluated the inoculation of potato with two arbuscular mycorrhizal fungi (AMF) strains, Claroideoglomus claroideum (CC), Claroideoglomus lamellosum (HMC26) and the MIX (CC+HMC26). Yield, phenolic profiles, and antioxidant activity at increasing levels of water stress. Result showed that the AMF did not show any influence on the number and biomass of the tubers. Inoculation with the individual AMF strains, especially C. lamellosum, resulted in a better response in terms of phenolic compounds and antioxidant activity measured as Oxygen Radical Absorbance Capacity (ORAC) in the potato tubers under water stress.

The overall research is well conducted and very information because this study suggesting specific AMF adaptations to water stress that must be considered when inoculation. In this sense, this manuscript is much more valuable. However, I found a lack of story connection and lack of potential references (some I suggested some below). Overall after I evaluate and request the author for this manuscript as a “MAJOR REVISION”.

Major Suggestions

  1. Abstract:Author should improve abstract further by novelty of the result and behalf of lengthening the methodology part. Author should rephrase the result and make it clear by addressing the common message for audiences. This is very important than only lengthening the text. Author should rephrase the abstract and whole manuscript should elucidate how this study is useful for the society?

R: Dear reviewer, thank you very much for your comments. Abstract was totally rewritten considering the indicated aspects, highlighting the main results and in the search of this information could be useful for the society.

  1. Introduction:Introduction is well presented the research objectives in the Ln 90-93 in the last section which is highly appreciated. However, please connect the research objectives with the research hypothesis. Author should clearly mention the research hypothesis, and this should relate to the research objectives. The hypothesis and research objectives should be very clear because, without appropriate literature, questions, or hypotheses in the introduction section the entire text will be unclear.

R: Thank you very much for your comments. The modifications were carried out in text.

  1. Concise the text:author should concise the text by removing the unnecessary and less important text. Please include the text related on the research title and its circumstances by cutback unnecessary text.

R: Thank you. Abstract and introduction were modified considering your comments.

Some line-to-Line comments

  1. Line no. 39 (Introduction):Author should mention increasing the world population and food hunger scenario and related data, which is much appreciated. Drought generally involves to reduction of morphological characteristics specially alter of leaf size and reduction of plant and leaf water contents, xylem sap flow and whole plant hydraulic conductivity because drought destruction and alter the anatomical characteristics refer this article https://doi.org/10.1016/j.foreco.2020.118099 for the further clarify the negative effect under drought in introduction.

R: The information was added in the introduction (L48-50).

  1. Line no. 226 (Discussion): Author should address why arises the antioxidant and secondary metabolites under drought? Please see the Ln 217-319 and Refer to these two articles for better clarify (1) https://doi.org/10.1038/s41598-019-55889 (2) https://doi.org/10.1016/j.scitotenv.2021.146466 and mention somewhere in that paragraph “abiotic stress especially environmental stress (e.g. UV radiation) plant produces the ROS when the plant exposed to the stress condition and plant produce antioxidant, flavonoids, and secondary metabolites , play to the role for protecting the plant for detoxifying ROS and protect the plant to protect from drought and help to stabilization of proline and amino acid”.

R: Thank you very much for the information. Unfortunately, the DOI indicated in (1) seems to have an error. Regarding the information available in (2), it was included in our manuscript (L289-293).

  1. Line no. 109/201 (Results):The legend should clearly describe about each figure. Accordingly, please check all the table and Figure legends throughout the manuscript. Moreover, the letter inside the figures is not clear in most of the figures please increase its font size and its visibility.

R: Thank you very much. We have checked the figures, and the legend of Figure 5 was modified. Also Figures 1-4 were modified based on your comments.

  1. Line no. 247-251 (Discussion):Author should further interpretation the negative effect of leaf morphological and growth traits and its related causes. Drought stress reduction of leaf water potential, chlorophyll content that reduction the physiological traits (photosynthesis) of the plant. “Generally, reduction of photosynthesis that effect on the growth and leaf morphological traits under drought”. Please refer this article https://doi.org/10.1016/j.scienta.2018.11.021 and elaborate the text more broadly.

R: The information was incorporated in the manuscript (L246-248).

  1. Line no. 437 (Conclusion): Conclusion should not be repetitive in the abstract or a summary of the results section. Please reduce the number of paragraphs which is generally not suited in the conclusion. Author should concise the text, I would love to read striking points and take-home messages that will linger in the readers’ minds. What is the novelty, how does the study elucidate some questions in this field, and the contributions the paper may offer to the scientific community?

R: Thank you. The conclusion was reduced and modified remarking the principal aspects.

  1. Line no. 467 (References): please include more related citation, check their pattern and writing style, spell check, and other grammatical errors. moreover, the author should cut the old and less matching literature and include the latest literature some of them are above.

R: Thank you very much for your comment. More related citation was added in introduction and discussion sections, based on all reviewers’ suggestions.

Round 2

Reviewer 1 Report

Comments and Suggestions for Authors

The manuscript is substantially improved and now can be corrected in its current form.

Author Response

Dear reviewer, thank you very much for your possitive appreciation about our study and manuscript. Best regards.

Reviewer 2 Report

Comments and Suggestions for Authors

The manuscript was improved after authors correction.

Author Response

Dear reviewer, thank you very much for your tpossitive appreciation about our study and manuscript

Reviewer 3 Report

Comments and Suggestions for Authors

Dear Author

I have read the revised manuscript plants-2749142. Entitled: Antioxidant Response, Phenolic Compounds and Yield of Solanum tuberosum Tubers Inoculated with Arbuscular Mycorrhizal Fungi and Growing Under Water Stress in plant MDPI. This is the second submission made by the author. The author addressed all the questions and suggestions that I raised the issue in the review. I satisfy the Author improved the abstract significantly. Author significantly improved their research hypothesis and well connected with the research objectives in this time. This manuscript improved the flow of writing, which was comparatively shallow in the original version but in this revised copy author very well addressed all the quarries and suggestions. Before accepting this manuscript, please check again the referencing. Further if there is anything needed to be revised by the author, especially English grammar, or spell check, I request this manuscript is currently in “Minor Revision” and the author may correct any further grammatical errors (if any) the author may improve in this stage.

Thank you.

Author Response

Dear reviewer, thank you very much again for your time and your comments and suggestions.

Effectively, the references were checked again, and some minor errors were corrected in the new version of the manuscript.

On the other hand, prior to submission, the English language of the manuscript was edited to improve its quality. The certificate proving said review is attached as pdf file
